# 24-hour movement behaviours and mental health in non-clinical populations: A systematic review

**Rachel Dale [1]\*, Teresa O'Rourke[1], Barbara Nussbaumer-Streit[2‡], Thomas Probst[1,3‡]**

**1** Department for Psychosomatic Medicine and Psychotherapy, University for Continuing Education Krems, Krems an der Donau, Austria, **2** Cochrane Austria, Department for Evidence-Based Medicine and Evaluation, University for Continuing Education Krems, Krems an der Donau, Austria, **3** Division of Psychotherapy, Department of Psychology, Paris Lodron University Salzburg, Salzburg, Austria

‡ Shared senior authorship.
\* rachel.dale@donau-uni.ac.at

## Abstract

The 24-hour movement guidelines consider movement behaviours (sleep, exercise, sedentary time) together within the frame of our 24-hour limit to provide recommendations on how a physically healthy day should look. There is increasing evidence that daily movement behaviours are associated with mental health. However the research into the relationship between 24-hour-movement and mental health, particularly in adults, is still to be systematically reviewed. The aim of this systematic review was to synthesise the current state of knowledge regarding movement behaviours and mental health in non-clinical child, adolescent and adult samples. systematic literature search of PubMed, Scopus and Embase was conducted in 2022, and updated in 2024. The review was preregistered (PROSPERO: CRD42022312717). Due to heterogeneity of methods and analyses, narrative synthesis of the results was employed. Of 103 eligible studies, one was a randomised controlled trial and the remainder were observational. In children 19/27 studies (70%) found at least one significant positive relationship between movement behaviour and mental health, in adolescents 38/41 (93%) and in adults 41/46 (89%). Certainty of evidence was low. More controlled studies are needed to make causal conclusions, but it is evident that the composition of movement behaviours is associated with mental health, and these associations may be differentially manifest in different age groups. This has implications for public health and mental health campaigns.

## 1. Introduction

The amount, type and quality of sleep, sedentary behaviour, low intensity and moderate-to-vigorous intensity physical activity a person partakes in is known to affect their health [1–3]. However, the typical lifestyle in a 21st century Western

**Data availability statement:** "All relevant data are within the paper and its Supporting Information files."

**Funding:** The author(s) received no specific funding for this work.

**Competing interests:** The authors have declared that no competing interests exist.

culture can make it difficult to move and sleep as much as our bodies are evolved to do [4] and due to advancements in food production and technology, many of us now lead predominantly sedentary lifestyles.

In light of the risks of sedentary habits, public health messages regarding movement and activity levels have become ubiquitous [5]. Building on this, the recent research [6–8] has highlighted that these activities, known together as movement behaviours, are mutually exclusive and they each have reciprocal effects on the others. Furthermore, we have a finite amount of time in each day. Consequently, increased time spent in one movement behaviour (e.g., sleep) must be compensated by an accompanying decrease in other movement behaviours (e.g., exercise). As such, their effects should not be examined in isolation, but rather they should be considered together. This has led to large-scale analyses of the effects of the composition of movement behaviours, i.e., light physical activity (LPA), moderate-vigorous physical activity (MVPA) muscle strengthening, sedentary behaviour, screen time (ST) and sleep, across the 24-hour day. This research was used to develop the 24-hour movement guidelines for children [9] and adults [8] to outline the ideal daily movement patterns for optimal physical health, thus providing a roadmap for counteracting the health risks of the sedentary nature of modern industrialised societies (Box 1).

Box 1.–Movement guidelines for adults and youth. Note: The guidelines summarised here are for adults 18–64 years and children 5–17 years. There are separate guidelines for >65 and <5 years. https://csepguidelines.ca/

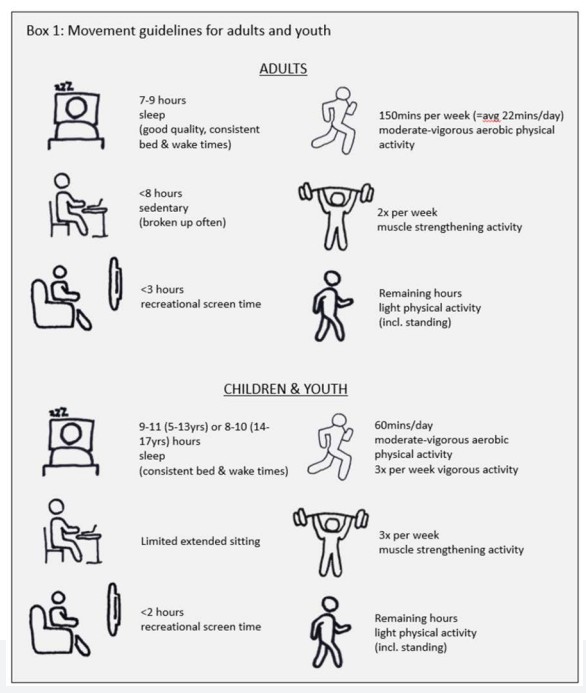

In order to assess movement across the day as a whole, compositional data analyses techniques (CoDA), whereby these behaviours across 24-hours can be encompassed in one composition variable [10], provides a valuable tool. This means that analyses on this composition adjusts for time spent in other behaviours rather than considering them in isolation. Building on this, isotemporal substitution models (ISM) can then be run on these compositions [11] to analyse the effect of substituting a certain amount of time spent in one behaviour, for example 15 minutes, for another behaviour whilst keeping the remaining compositions constant. This can shed light on which activities or daily composition of activities should be prioritised for the desired health outcome (e.g., reduced chance of cardiac problems). Other studies take the guidelines as the measure of interest and analyse the health associations of meeting some, all, or various combinations of, the recommended 24-hour movement guidelines with health outcomes. Henceforth guidelines refer to the overall 24-hour movement guidelines, and recommendations refer to the individual recommendations within those guidelines (e.g., meeting the physical activity recommendation).

Previous reviews have used these methods to assess the association between 24-hour movement and general health outcomes [6,7,12,13]. Mental health, however, played a minimal role in the development of the 24-hour movement guidelines [7]. Nonetheless it is well-established through reviews and meta-analyses that the individual movement behaviours, when considered separately, do have significant effects on bolstering mental well-being and reducing mental illness [14–19]. For example, meta-analyses have revealed physical activity reduces depression and anxiety symptoms [14,15,18], reduced sedentary time lowers anxiety and increases wellbeing [16,17] and better sleep improves a number of mental health measures [19]. Therefore, there is reason to believe that our movement behaviours as a whole likely also have an effect on mental health.

Since the publication of the 24-hour movement guidelines, a number of researchers have now begun to investigate the relationships between movement behaviour composition and/or guideline adherence and mental health, as evidenced in our results below. However, studies have sometimes produced seemingly disparate results. Furthermore, with so many behaviours and variables to take into account, it is challenging to build a full picture of how daily movement and mental health are connected. For example, most studies do not consider all possible movement behaviours (LPA, MVPA, muscle strengthening, sleep, sedentary time, screen time) but rather select a subset. Furthermore, the relationship between movement and mental health may depend on the population in question and in non-clinical studies the focus can be on different sub-samples of the general population (e.g., office workers [20], university students [21]). Therefore, systematic reviews of how the associations of movement behaviours with mental health hold up across populations, behaviours measured, and analysis techniques employed are necessary.

Three systematic reviews so far have investigated how combinations of physical activity, sedentary time and sleep duration relate to mental health indicators specifically in children and adolescents [13,22,23]. Across 10 cross-sectional studies included in the review by Sampasa-Kanyinga et al. [22], better mental health among children and youth was observed among those who met all three recommendations as compared with those who did not meet recommendations. Furthermore, a dose-response gradient was also observed. Just three years later a similar systematic review included 30 studies [23]. Eight of 12 studies found an association between movement and socioemotional outcomes, all nine studies found an association with health-related quality of life and 7/12 found an association with depression or anxiety. The authors highlight the roles of lower sedentary time and high sleep for psychological health. Finally Zhao et al [13] found 8/13 studies to show an association between meeting all guidelines and better mental health, echoing findings by Sampasa-Kanyinga et al. [22]. A more recent scoping review on children and youth included 55 articles and found an association between mental health and sleep in ~70% of articles, ~60% with physical activity and 60–70% with sedentary behaviour [24].

However several studies have been published since these reviews and, importantly, it is clear the reviews so far only included younger populations. There is yet to be a review focussing on the associations with mental health outcomes including adult populations. Therefore, this current review aimed to synthesise the current state of knowledge

in non-clinical samples of children, adolescents and adults to assess commonalities or disparities in findings to date regarding associations between 24-hour movement and mental health across populations, designs and analysis techniques. Furthermore we aimed to identify gaps in the literature where more research is especially needed. It was hypothesised that movement behaviour compositions more in line with the 24-hour movement guidelines would be positively associated with mental health outcomes in children, adolescents and adults, and across analysis techniques.

## 2. Methods

### 2.1 Registration and protocol

This systematic review was registered a priori with Prospective Register of Systematic Reviews (PROSPERO registration number: CRD42022312717, [25]). The review was conducted in accordance with the Preferred Reporting Items for Systematic Reviews and Meta-Analyses (PRISMA) statement [26] (Fig 1, S3 File).

### 2.2 Eligibility criteria

The PICOS framework was used to identify a priori study features [27].

**PRISMA 2020 flow diagram for new systematic reviews which included searches of databases, registers and other sources**

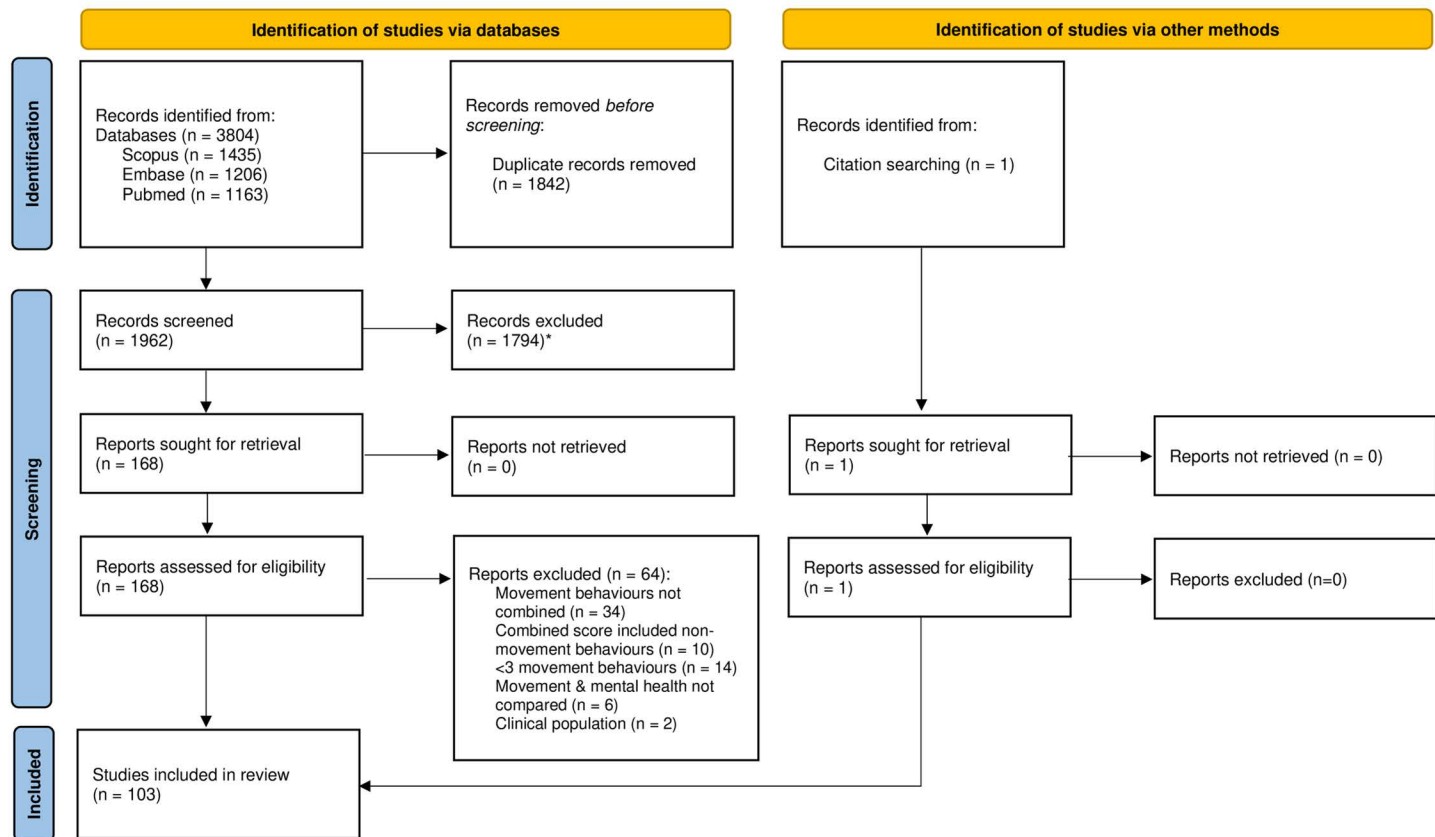

**Fig 1. Flow diagram of article screening.** * Reasons: excluded population n = 234, < 3 movement behaviours n = 375, movement behaviours not combined n = 134, other reasons n = 2.

**2.2.1 Population.** The population of interest for the current review was non-clinical, general populations of children, adolescents, or adults. The age range definitions can vary slightly from study to study (supplementary materials) however typically children were 12 or under, adolescents 13–18 and adults >18 years.

The population did not include people with diagnosed serious physical or psychological conditions requiring acute/hospital care. Also excluded were populations with special movement requirements such as samples with paraplegia or elite athletes.

**2.2.2 Intervention/exposure.** The intervention/exposure of focus was the composition of time across the 24-hour day spent in any three or more of the following movement behaviours; sleep, sedentary, screen time, muscle building, LPA and MVPA. Alternatively, the focus could be whether or not participants met the recommendations [7,9] for three or more movement behaviours. The movement behaviours could be assessed objectively or via self-report. These movement behaviours are based on the definitions used by the individual studies (see supplementary materials).

Exclusion criteria: studies which assessed two or fewer movement behaviours.

**2.2.3 Comparison/control.** The comparator was the composition of movement behaviours or the movement behaviour recommendations met. Specifically, the composition of time spent in sleep, sedentary/screen time, LPA or MVPA *relative to the other behaviours*, or different *combinations of* recommendations met. Therefore studies including individual behaviours or recommendations, without accounting for the others, were not included. Changes in movement behaviour composition were also considered for longitudinal studies, intervention studies or studies which modelled reallocation of time from one behaviour to another.

**2.2.4 Outcome.** The outcomes were any mental illness symptoms or mental well-being measures; common examples include anxiety, depression, quality of life and well-being. No limits were placed on which specific measures were used.

**2.2.5 Study designs.** Included studies were controlled intervention studies (randomised controlled trial, non-randomised controlled trial) as well as observational studies with a control condition and cross-sectional, prospective and retrospective designs. Neither case studies nor systematic reviews with or without meta-analyses were included. Studies needed to use some method of accounting for the proportion of the day spent in each movement behaviour, such as compositional data analysis (CoDA), isotemporal substitution analysis (ISM) or accounting for different combinations of recommendations met by participants in the analyses using latent class analysis or similar.

### 2.3 Information sources and search strategy

Studies were eligible if they were published in peer-reviewed journals in English or German. No limit was placed on the year of publication. Three databases were searched [28]: PubMed, Scopus and Embase, using the following summarised search terms:

1. "24-hour movement" OR "24-hr movement" OR "movement behaviour" OR "movement guidelines" OR ("sleep" AND "sedentary" AND "activity")

AND

2. "mental health" OR "depression" OR "anxiety" OR "wellbeing" OR "psychosocial functioning" OR "emotions"

The full search strategies were checked by an information specialist and can be found in the supplementary materials (S1 Table). The search strategy was based on a previous study in the field [6] and the 'movement' block of the search finds all those included in their review, confirming the performance of this part of the search. References of identified studies and references of existing similar reviews were also checked. Searches were conducted on 15/02/2022 and updated searches were conducted on 26/08/2024.

## 2.4  Study selection

After searching the aforementioned databases, references were exported to a reference manager and duplicates were removed. Titles and abstracts were then screened by RD and TO independently in the reference managers and those not meeting the inclusion criteria were removed. The same reviewers then examined the full texts of the remaining 168 articles dually and independently (Fig 1).

## 2.5  Data extraction

Data were extracted from eligible articles into Microsoft Excel. Two authors extracted the data from the same five articles, the remainder were extracted by RD and checked by TO. The basic information of the study was extracted: design, population studied, sample size, average age, percentage of each gender/sex. The movement behaviours measured, and method of measurement were also extracted. Likewise, mental health measures and method used were entered. Finally, any covariates accounted for in the analyses and the analysis technique were also recorded. For studies which reported results from multiple models or analyses, only the results from the most fully adjusted models were extracted.

## 2.6  Risk of bias and study quality assessment

Risk of bias was assessed for articles by two authors (RD & TO) using accepted tools: the Cochrane RoB2 tool for randomised controlled trials [29], the Joanna Briggs Institute checklist for cross-sectional studies [30] and the Newcastle Ottawa quality assessment scale for cohort studies [31]. If an article used more than one of the above designs, both risk of bias assessments were conducted. The certainty of evidence was assessed for the six outcomes most commonly assessed in mental health research (depression, anxiety, general mental health, emotional problems, quality of life and stress, [32]), with the GRADE approach [33], using a framework for reviews when pooled effect estimates are not possible to obtain [34]. The detailed risk of bias and certainty of evidence assessments for each study/outcome can be seen in S2 File.

## 2.7  Synthesis and interpretation of results

The studies varied greatly in population sampled, design, method of measuring and analysing movement behaviours, outcomes assessed and analyses employed to compare movement behaviours with outcome (Tables 1 and 2). This presented a challenge for data synthesis and prevented a meta-analysis from being conducted. Since most of the studies were observational and without comparison conditions, standardised means were not calculated. The studies are presented according to age group (children, adolescents and adults) and within that they are divided according to outcome measured and then analysis category employed. Data were synthesised for all studies meeting the inclusion criteria.

## 3.  Results

### 3.1  Description of the studies

After systematic searches of PubMed, Scopus and Embase, 3804 records were identified of which 1962 remained after removal of duplicates. After abstract and full-text screening 103 articles were eligible for inclusion (Fig 1).

Characteristics of the 103 included studies are summarised in Tables 1 (children and adolescents) and 2 (adults). The articles were published between 2014 and 2024 and the sample sizes ranged from 73–238,440. Twenty-seven studies included children (<12 years), 41 included adolescents and 46 were with adults. Some studies included both children and adolescents (Table 1), but the results are presented separately for the two age groups. The risk of bias and quality of evidence grading can be seen in S2 File. Due to the observational nature of most studies, the

**Table 1. Description of included studies: children (green) and adolescents (yellow).**

| Reference | Design | Popula-tion | Sample size | Mean age | % female | Sleep | Seden-tary | Total PA | LPA | MVPA | Screen time | Strength training | Outcomes |
|---|---|---|---|---|---|---|---|---|---|---|---|---|---|
| | | | | | | **Movement behaviours** | | | | | | | |
| Bang et al 2020 | Cross-sectional | 5-11-years Canada / 12-17-years | 4250 | NR | 48.7 | S-R/P-R | | | | accel | S-R/P-R | | SDQ, 1-item stress, 1-item mental health |
| Bao et al 2024 | Cross-sectional | 9-12-years China | 2005 | 9.83 | 48.4 | S-R | | | | S-R | S-R | | SDQ |
| Brown et al 2021c | Cross-sectional | 4-5-years Canada | 589 | 4.9 | 42.6 | | accel | | accel | accel | | | CBC |
| Carson et al 2019 | Cross-sectional | 3-years Canada | 343 | 3 | 47.9 | accel | | | accel | accel | P-R | | CBC |
| Fairclough et al 2021 | Cross-sectional | 9-11-years England / 12-13-years | 359 | 11.5 | 50.7 | accel | | | accel | accel | | | RSES, MFQ, SDQ |
| Fairclough et al 2023 | Cross-sectional | 9-11-years England | 301 | 11.1 | 60.1 | accel | accel | | accel | accel | | | SDQ |
| Fung et al 2023 | Cross-sectional & longitudinal | 9-14-years USA | 9273 | 9.9 | 48 | S-R/P-R | | S-R/P-R | | | YSTS | | CBC |
| Hansen et al 2022 | Cross-sectional | 9-12-years Germany / 13-18-years | 15786 | 13 | 50 | S-R | | | | S-R | S-R | | SDQ- depression subscale |
| Hinkley et al 2020 | Longitudinal | 3-5-years Australia | 98 | 4.6 | 47 | P-R | | | accel | accel | | P-R | SDQ, PedsQL |
| Hou et al 2023 | Cross-sectional | 6-12-years USA / 13-17-years | 907 | 12.22 | 54.58 | P-R | | | | P-R | P-R | | Depression diagnosis, resilience 1-item |
| Hou et al 2024 | Cross-sectional | 6-12-years USA / 13-17-years | 6030 | 12.79 | 55.7 | P-R | | | | P-R | P-R | | Internalising 2-items, externalising 6-items |
| Kasai et al 2024 | Cross-sectional | 6-12-years Japan | 2660 | NR | 49.29 | S-R | | | | S-R | S-R | | Jikaku-sho shirabe |
| Kuzik et al 2020 | Cross-sectional | 3-5-years Canada | 95 | 4.5 | 30.5 | accel | accel | | accel | accel | | | CSBQ |
| Li et al 2024 | Cross-sectional | 3-6-years China | 205 | 4.8 | 42.92 | P-R | accel | | accel | accel | P-R | | SDQ |

*(Continued)*

| Reference | Design | Popula-tion | Sample size | Mean age | % female | Movement behaviours | | | | | | | Outcomes |
|---|---|---|---|---|---|---|---|---|---|---|---|---|---|
| | | | | | | Sleep | Seden-tary | Total PA | LPA | MVPA | Screen time | Strength training | |
| Liang et al 2023 | Cross-sectional | 10-12-years China | 67821 | 13 | 48.1 | S-R | | | | S-R | S-R | | WHO5, CD-RISC-10, PHQ-9, GAD-7 |
| | | 13-17-years | | | | | | | | | | | |
| Lopez-Gil et al 2022 | Cross-sectional | 4-5-years 6-12-years Spain | 3772 | 9.5 | 49.4 | P-R | | | | P-R | P-R | | SDQ |
| | | 13-14-years | | | | | | | | | | | |
| McNeill et al 2020 | Cross-sectional & longitudinal | 3-5-years Australia | 185 | 4.2 | 40.1 | P-R | | accel | | accel | P-R | | SDQ |
| Peralta et al 2022 | Longitudinal | 5-12-years Switzer-land | 2534 | NR | 51.5 | P-R S-R | | | | P-R S-R | P-R S-R | | CL |
| | | 13-16-years | | | | | | | | | | | |
| Rorem et al 2024 | Cross-sectional & longitudinal | 3-5-years Canada | 375 | 3 5 | 49.6 | accel | accel | | accel | accel | CTVQ | | CBC |
| Sampasa-Kanyinga et al 2021c | Cross-sectional | 9-11-years USA | 11875 | 9.9 | 47.9 | P-R | | | | S-R | YSTS | | CBC |
| Sun et al 2023 | Cross-sectional | School age China | 1098 | 11.6 | 48.5 | S-R | | | | S-R | S-R | | WHO5 |
| Sun et al 2024 | Cross-sectional | 6-12-years USA | 2800 | 12.35 | 47.83 | S-R | | | | S-R | S-R | | Emotional prob-lems 4-items |
| | | 13-17-years | | | | | | | | | | | |
| Tan et al 2023 | Cross-sectional & longitudinal | 8-10-years Singa-pore | 370 | 8 10 | 50.5 | accel | accel | | accel | accel | | | KINDL-Kid |
| Taylor et al 2021 | Longitudinal | 5-years New Zealand | 528 | 5 | 48.5 | accel | accel | accel | | | P-R | | BASC-2 |
| Yin et al 2024 | Cross-sectional | 3-6-years China | 205 | 4.8 | 42.93 | CSHQ | | | | accel | P-R | | SDQ |
| Zhu et al 2019 | Cross-sectional | 6-11-years USA | 35718 | 11.5 | 49 | P-R | | | | P-R | P-R | | Diagnosed anxiety/depression |
| | | 12-17-years | | | | | | | | | | | |
| Zhu et al 2023 | Cross-sectional | 3-6-years China | 200 | 4.79 | 49 | P-R | | | | accel | P-R | | SDQ |

*(Continued)*

| Reference | Design | Popula-tion | Sample size | Mean age | % female | Movement behaviours | | | | | | | Outcomes |
|---|---|---|---|---|---|---|---|---|---|---|---|---|---|
| | | | | | | Sleep | Seden-tary | Total PA | LPA | MVPA | Screen time | Strength training | |
| Brown & Kwan 2021 | Cross-sectional | 14-17-years Canada | 1118 | 15.9 | 54.5 | S-R | | | | IPAQ | S-R | | FS, RSES. 2-items resiliency |
| Brown et al 2021a | Cross-sectional | 14-17-years Canada | 1253 | 15.9 | 54 | S-R | | | | IPAQ | S-R | | FS, RSES. 2-items resiliency |
| Brown et al 2021b | Cross-sectional | 12-18-years USA | 6436 | 16.03 | 51.7 | S-R | | | | S-R | S-R | | CES-D |
| Burns et al 2020 | Cross-sectional | 14-19-years USA | 1849 | 15.8 | 50.3 | S-R | | | | S-R | S-R | S-R | Perceived lonliness 1-item, prolonged sadness 1-item |
| Cao et al 2020 | Cross-sectional | 12-16-years China | 4178 | 14.25 | 53.4 | S-R | | | | S-R | S-R | | CES-D |
| Chong et al 2021 | Cross-sectional & longitudinal | 10-13-years Australia | 88 | 12.8 | 59.1 | accel | accel | | accel | accel | ASAQ | | SDQ, K-10 |
| Dumuid et al 2021 | Cross-sectional | 11-12-years Australia | 1182 | 12 | 49 | accel | accel | | accel | accel | | | PedsQL, MFQ, SDQ, BMSLSS |
| Duncan et al 2022 | Cross-sectional | 14-18-years Canada | 2645 | NR | 64.4 | S-R | | | | S-R | S-R | | CES-D, GAD-7, FS, emotional dys-regulation 6-itms |
| Duncan et al 2024 | Cross-sectional | 17-18-years Canada | 67248 | NR | 51.2 | S-R | | | | S-R | S-R | | FS, GAD-7, CES-D, BRS |
| de Faria et al 2022 | Cross-sectional | 15-17-years Brazil | 217 | 16 | 49.3 | accel | accel | | accel | accel | | | GHQ-12 |
| Faulkner et al 2020 | Longitudinal | 14-18-years Canada | 2292 | 16.3 | 53.6 | S-R | | | | SHAPES | S-R | | FS |
| Gilchrist et al 2021 | Cross-sectional | 14-18-years Canada | 46413 | NR | 51.5 | S-R | | | | SHAPES | S-R | | CES-D, FS |
| Huang et al 2024 | Cross-sectional | 12-16-years China | 15071 | 14.53 | 48.1 | PSQI | | | | PAQ-A | YSTS | | SDQ |
| Janssen et al 2017 | Cross-sectional | 10-17-years Canada | 21821 | NR | 52.8 | S-R | | | | S-R | S-R | | CL, emotional problems 9-items |
| Khan et al 2021 | Cross-sectional | 12-13-years Australia | 3096 | 12.37 | 48.7 | S-R | | | | S-R | P-R | | PedsQL |
| Khan et al 2024 | Cross-sectional | 13-17-years Bangla-desh | 312 | 14.31 | 42 | S-R | | | | IPAQ | ASAQ | | CES-D |
| Loewen et al 2019 | Cross-sectional | 10-11-years Canada | 3436 | NR | 51 | S-R | | | | S-R | S-R | | Diagnosis of men-tal illness |

*(Continued)*

| | | | | | | Movement behaviours | | | | | | | |
|---|---|---|---|---|---|---|---|---|---|---|---|---|---|
| Reference | Design | Popula-tion | Sample size | Mean age | % female | Sleep | Seden-tary | Total PA | LPA | MVPA | Screen time | Strength training | Outcomes |
| Lopez-Gil et al 2024 | Cross-sectional | 12-17-years USA | 44734 | NR | 48.55 | S-R | | | | S-R | S-R | | Suicidality 3-items |
| Lu et al 2021 | Cross-sectional | 10-13-years China | 5357 | 11.5 | 44.4 | S-R | | | | S-R | S-R | | PHQ-9, GAD-7 |
| Luo et al 2023 | Cross-sectional | 13-15-years China | 9420 | 14.53 | 55.19 | S-R | | | | S-R | S-R | | Anxiety 3-items, depression 6-items |
| Monteagudo et al 2023 | Longitudinal | 13-16-years Spain | 197 | 13.9 | 46.2 | | accel | | accel | accel | | | BASC |
| Patte et al 2020 | Longitudinal | 14-18-years Canada | 2292 | NR | 53.6 | S-R | | | | S-R | S-R | | CES-D |
| Sampasa-Kanyinga et al 2021a | Longitudinal | 14-18-years Canada | 14620 | 14.9 | 46 | S-R | | | | S-R | S-R | | CES-D |
| Sampasa-Kanyinga et al 2021b | Cross-sectional | 12-18-years Canada | 6364 | 15.1 | 48.3 | S-R | | | | S-R | S-R | | K-6 |
| Sampasa-Kanyinga et al 2022a | Cross-sectional | 11-20-years Canada | 12699 | 15 | 51 | S-R | | | | S-R | S-R | | Mental health 1-item |
| Sampasa-Kanyinga et al 2022b | Cross-sectional | 11-20-years Canada | 6932 | 15.2 | 56.77 | S-R | | | | S-R | S-R | | Stress 1-item, self-esteem 1-item |
| Yuan et al 2023 | Cross-sectional | 13-18-years Inner Mongolia | 238440 | NR | 50.5 | S-R | | | | S-R | S-R | | CES-D |
| Zhang et al 2023a | Cross-sectional & longitudinal | 12-13-years China | 906 | NR | 49 | S-R | | | | S-R | S-R | | GAD-7, PHQ-9 |
| Zhang et al 2023b | Longitudinal | 12-18-years China | 816 | 14.76 | 51 | PSQI | | | | IPAQ | ASAQ | | WHO5 |
| Zhou et al 2024 | Cross-sectional | 12-16-years China | 670 | 13.57 | 42.8 | S-R | S-R | | | S-R | | | SAS |

Note: accel = accelerometer, ASAQ = adolescent sedentary activity questionnaire, BASC = behavioural assessment system for children, BMSLSS = brief multidimensional students' life satisfaction scale, BRS = brief resiliency scale, CBC = child behavior checklist, CD-RISC-10 = Connor-Davidson resilience scale, CES-D = centre of epidemiologic studies depression scale, CL = cantril ladder, CSBQ = child self-regulation and behaviour questionnaire, CSHQ = children's sleep habits questionnaire, CTVQ = childhood television viewing habits questionnaire, FS = flourishing scale, GAD-7 = generalized anxiety disorder scale, GHQ-12 = general health questionnaire, IPAQ = international physical activity questionnaire, K-10/K-6 = kessler's psychological distress scale, MFQ = mood and feelings questionnaire, NR = not reported, PAQ-A = physical activity questionnaire for adolescents, PedsQL = pediatric quality of life inventory, PHQ-9 = patient health questionnaire, RSES = rosenberg self-esteem scale, SAS = Zung's self-rating anxiety scale, SDQ = strengths and difficulties questionnaire, SHAPES = school health action, planning and evaluation system physical activity questionnaire, S-R = self-report, P-R = parent-report, YSTS = youth screen time survey.

S-R and P-R denote single and/or unvalidated items.

**Table 2. Description of included studies: adults.**

| Reference | Design | Population | Sample size | Mean age | % female | Movement behaviours | | | | | | | Outcomes |
|---|---|---|---|---|---|---|---|---|---|---|---|---|---|
| | | | | | | Sleep | Sedentary | Total PA | LPA | MVPA | Screen time | Strength training | |
| Baillot et al 2022 | Cross-sectional | General population Canada | 10515 | 45.4 | 50.3 | S-R | S-R | | | accel | S-R | | Mental health 1-item |
| Blodgett et al 2023 | Cross-sectional | General population UK | 4738 | 46 | 52.3 | accel | accel | | accel | accel | | | Medication/doctor's visit |
| Brown et al 2022a | Cross-sectional | Young adults Canada | 15080 | 20.78 | 67.1 | S-R | | | | IPAQ | S-R | | K-10, WEMWBS |
| Brown et al 2022b | Cross-sectional | Young adults Canada | 17633 | 21.7 | 67.1 | S-R | | | | IPAQ | S-R | | Suicidality 2-items |
| Bu et al 2021 | Cross-sectional | Students China | 1846 | 20.7 | 64 | PSQI | IPAQ | | | IPAQ | | | SAS |
| Cabanas-Sanchez et al 2021 | Cross-sectional & longitudinal | Over 65s Spain | 1679 | 71.7 | 53.1 | accel | accel | | accel | accel | | | CL, GDS-10, LS, SF-12 |
| Chao et al 2022 | Cross-sectional | Students China | 1475 | 20.7 | 68 | PSQI | IPAQ | | | IPAQ | | | SAS |
| Colley et al 2018 | Cross-sectional | General population Canada | 10621 | 45.3 | 52 | S-R | accel | | accel | accel | | | Mental health 1-item |
| Curtis et al 2020 | Cross-sectional | Less active Australia | 430 | 41.3 | 74 | accel | accel | | accel | accel | | | SF-12, DASS-21 |
| del Pozo Cruz et al 2020 | Cross-sectional | Representative USA | 3233 | 47.4 | 52.1 | SDQ | accel | | accel | accel | | | PHQ-9 |
| Dennis et al 2021 | Cross-sectional | Preconception & recently pregnant parents Canada | 1304 | 34.7 | 82.8 | S-R | S-R | | | GPAQ | S-R | | PHQ-9, GAD-7 |
| Duncan et al 2021 | RCT | Inactive, poor sleep Australia | 160 | 41.5 | 80 | PSQI | WSQ | | | AAQ | | S-R | DASS-21, SF-12 |
| Feng et al 2022a | Cross-sectional | Pre-school caregivers China | 2002 | 35.5 | 76.3 | S-R | | | | IPAQ | S-R | | DASS-21 |
| Feng et al 2022b | Longitudinal | Students China | 410 | 19.3 | 58.8 | PSQI | IPAQ | | | IPAQ | S-R | | PCL-C |
| García-Hermoso et al 2022 | Longitudinal | General population USA | 7069 | 15.35 | 56.8 | S-R | | | | S-R | S-R | | CES-D, suicidality 1-item |
| Guallar-Castillon et al 2014 | Longitudinal | General population Spain | 4887 | 54.3 | 50.8 | S-R | NHS | | | EPIC | | | SF-12 |

*(Continued)*

| Reference | Design | Popula-tion | Sam-ple size | Mean age | % female | Sleep | Seden-tary | Total PA | LPA | MVPA | Screen time | Strength training | Outcomes |
|---|---|---|---|---|---|---|---|---|---|---|---|---|---|
| | | | | | | **Movement behaviours** | | | | | | | |
| Haegele et al 2021 | Cross-sectional | Visual impair-ments USA | 182 | 44.8 | 65.4 | S-R | IPAQ | | | IPAQ | | | MDI |
| Hajo et al 2020 | Cross-sectional | Nurses Canada | 342 | 43.1 | 94 | S-R | accel | | | accel | | | SWDSQ, POMS |
| Hidde et al 2022 | Cross-sectional | Cancer survivors USA | 73 | 53 | 75.7 | accel | accel | | accel | accel | | | FACT |
| Hofman et al 2021 | Cross-sectional | Over 45s Nether-lands | 1943 | 71 | 52 | accel | accel | | accel | accel | | | CES-D, HADS |
| Jiang et al 2024 | Cross-sectional | Over 60s China | 648 | 72.65 | 60.6 | PSQI | S-R | IPAQ | | | | | GDS-15 |
| Kandola et al 2021 | Longitudinal | 40-69-years UK | 60235 | 55.9 | 56 | S-R | accel | | accel | accel | | | PHQ-9, GAD-7 |
| Kastelic et al 2021 | Cross-sectional | General population Slovenia | 2333 | 48 | 74 | DABQ | DABQ | | | DABQ | | | Stress 1-item |
| Kitano et al 2020 | Cross-sectional | Office workers Japan | 1095 | 50.2 | 68.6 | S-R | accel | | accel | accel | | | K-6, UWES |
| Kostick & Zhu 2024 | Cross-sectional | Priests USA | 335 | 51.61 | 0 | S-R | | | | IPAQ | S-R | | HADS |
| Larisch et al 2020 | Cross-sectional | Office workers Sweden | 662 | 41 | 68 | accel | accel | | accel | accel | | | HADS, SMBM, WHO-5, stress 1-item |
| Le et al 2021 | Cross-sectional | General population Australia | 361 | 22.5 | 66.8 | accel | accel | | accel | accel | | | PNAS |
| Liang et al 2021 | Cross-sectional | Students China | 1846 | 20.67 | 64 | PSQI | IPAQ | | IPAQ | IPAQ | | | PHQ-9, SAS |
| Liang et al 2024a | Cross-sectional | 60-79-years China | 4562 | 67.68 | 55.8 | IPAQ | IPAQ | | | IPAQ | | | PHQ-9, ESLS-10 |
| Liang et al 2024b | Cross-sectional | 60-79-years China | 4562 | 67.68 | 55.8 | IPAQ | IPAQ | | IPAQ | IPAQ | | | PHQ-9, ESLS-10 |
| Lin et al 2024 | Cross-sectional | Young adults China | 1742 | 20.03 | 68.6 | S-R | | | | IPAQ | SBQ | S-R | DASS-21, WHOQOL |
| Liu et al 2023 | Cross-sectional | Office workers China | 10656 | 33.06 | 55.6 | S-R | IPAQ | IPAQ | | | | | CES-D |
| Liu et al 2024 | Cross-sectional | General population USA | 2803 | 48.03 | 50.8 | S-R | accel | | accel | accel | | | PHQ-9 |
| Luo et al 2022 | Cross-sectional | 60-80-years China | 4134 | 67.37 | 53.12 | S-R | S-R | S-R | | | S-R | | Mental health 3-items |

*(Continued)*

| Reference | Design | Popula-tion | Sam-ple size | Mean age | % female | Movement behaviours | | | | | | | Outcomes |
|---|---|---|---|---|---|---|---|---|---|---|---|---|---|
| | | | | | | Sleep | Seden-tary | Total PA | LPA | MVPA | Screen time | Strength training | |
| McGregor et al 2018 | Cross-sectional | 18-64-years 65-79-years Canada | 6322 & 1454 | 41.3 & 69.3 | 49.6 & 52.4 | S-R | accel | | accel | accel | | | Mental health 1-item |
| Meneguci et al 2024 | Cross-sectional | Over 60s Brazil | 473 | 70.2 | 62.6 | S-R | IPAQ | | | IPAQ | | | GDS-10 |
| Meyer et al 2020 | Cross-sectional & longitudinal | Students USA | 423 | 27.6 | 50 | accel | accel | | accel | accel | | | POMS, PSS-10 |
| Murray et al 2023 | Cross-sectional | Young adults Canada | 770 | 20.4 | 55 | S-R | S-R | | IPAQ | IPAQ | | | MDI, mental health 1-item |
| Ohta et al 2023 | Cross-sectional | General population Japan | 640 | 64.1 | 58.3 | S-R | GPAQ | | | GPAQ | | | CES-D |
| Perez et al 2021 | Cross-sectional | Military USA | 17166 | NR | 16.7 | S-R | | | | S-R | S-R | | PC-PTSD-5, suicidal ideation 1-item, K-6 |
| Shi et al 2024 | Longitudinal | 50-70-years China | 45176 | 59.2 | 100 | S-R | S-R | | S-R | S-R | S-R | | GDS-15 |
| Su et al 2022 | Cross-sectional | Students China | 1475 | 20.7 | 68 | S-R | IPAQ | | IPAQ | IPAQ | | | PHQ-9 |
| Tabaczynski et al 2020 | Cross-sectional | Cancer survivors Canada | 463 | 62.7 | 36.1 | S-R | STQ | | GLTQ | GLTQ | | | FACT |
| Wang et al 2023 | Longitudinal | Students China | 437 | 20.1 | 51.7 | PSQI | SBQ | | IPAQ | IPAQ | | | DASS-21 |
| Weatherson et al 2021 | Cross-sectional | Students Canada | 20090 | 24.1 | 67 | S-R | S-R | | IPAQ | | S-R | | K-10, WEMWBS |
| Zhang et al 2024 | Cross-sectional | Students China | 1793 | 20.7 | 63.6 | PSQI | IPAQ | | | IPAQ | | | PHQ-9 |
| Zhu et al 2024 | Longitudinal | General population UK | 84168 | 56.2 | 55.4 | accel | accel | | accel | accel | | | Depression diagnosis |

Note: AAQ = active Australia questionnaire, accel = accelerometer, CES-D = centre of epidemiologic studies depression scale, CL = cantril ladder, DABQ = daily activity behaviours questionnaire, DASS-21 = depression anxiety stress scale, EPIC = European prospective investigation into cancer and nutrition physical activity questionnaire, ESLS-10 = emotional and social loneliness scale, FACT = functional assessment of cancer therapy scales, GAD-7 = generalized anxiety disorder scale, GDS = geriatric depression scale, GLTQ = godin leisure-time questionnaire, GPAQ = global physical activity questionnaire, HADS = hospital anxiety and depression scale, IPAQ = international physical activity questionnaire, K-6/K-10 = kessler's psychological distress scale, LS = loneliness scale, MDI = major depression inventory, NHS = nurses' health study, NR = not reported, PCL-C = posttraumatic-stress disorder checklist-civilian version, PC-PTSD-5 = primary care PTSD screen for DSM-5, PHQ-9 = patient health questionnaire, PNAS = positive and negative affect schedule scales, POMS = profile of mood states questionnaire, PSQI = pittsburgh sleep quality index, PSS-10 = perceived stress scale, RCT = randomised controlled trial, SAS = Zung's self-rating anxiety scale, SBQ = sedentary behavioral questionnaire, SDQ = sleep disorder questionnaire, SF-12 = short form health survey, SMBM = shirom-melamed burnout measure, S-R = self-report, STQ = domain specific sitting time questionnaire, SWDSQ = shift work disorder screening questionnaire, UWES = Utrecht work engagement scale, WEMWBS = warwick-edinburgh mental well-being scale, WHO-5 = world health organisation five well-being index, WSC = workforce sitting questionnaire

S-R denotes single or unvalidated items.

quality was rather low. Of cross-sectional and longitudinal studies, the most common issue was failing to measure all exposures (movement behaviours) in a valid and reliable way: only 35% did so for all behaviours in cross-sectional studies and 44% for longitudinal studies. However, it should be noted 78% measured the outcome in a valid and reliable way.

Most studies used cross-sectional data (n = 87), 23 studies were longitudinal (seven of which conducted both cross-sectional and longitudinal analyses) and one was a randomised controlled trial. Follow-up durations for longitudinal studies varied from 2 months to 24 years, with the mode being 1 year (S4 File). Many studies (n = 35) used accelerometers to objectively measure one or more movement behaviours, some used validated questionnaires, but unvalidated self- or parent-report items were also widely used. Most studies used validated questionnaires or clinical diagnoses to measure the mental health outcomes but 19 studies used unvalidated items for some or all of the outcome measures. A range of analysis techniques were also employed, resulting in high heterogeneity of the data (CoDA, ISM, grouping analyses; see S2-S4 Tables in S1 File). More details on the characteristics can be seen in the supplementary materials.

## 3.2 Data synthesis

The results are presented separately for children, adolescents and adults. Within these sub-groups, summaries of the findings for each outcome, according to analysis category, are presented in S2-S4 Tables in S1 File and further summarised in Tables 3–5. A comparison of summarised findings for each age group can be seen in Fig 2. Overall, at least one significant, positive association between movement behaviours and mental health was found in 19/27 studies with children (70%), 38/41 studies with adolescents (93%) and 41/46 studies with adults (89%). Certainty of evidence was very low for all assessed outcomes (S2 File).

**3.2.1 Children.** *Emotional problems:* Emotional problems include the broad constructs of internalising problems (emotional problems and peer relationships) and externalising problems (behavioural problems and hyperactivity, e.g., [35]). Of 18 studies, 12 found an association of movement behaviours with emotional problems. Meeting more recommendations (especially screen time) was associated with lower levels of emotional, internalising, or externalising problems in 8/12 studies [36–43]. Four of five CoDA studies found that more sedentary/screen time was associated with more emotional problems [35,44–46], with the other finding no associations [47], and two of four ISM studies found replacing sleep or MVPA with sedentary/screen time resulted in higher emotional problems [44,46]. However three studies found an association between high MVPA and externalising problems [38,42,46].

*Depression:* All six studies found that meeting any recommendation (ST, MVPA, sleep, sedentary) and meeting three recommendations was associated with lower depression in children [41,48–52]. Meanwhile the study employing CoDA did not find an association between movement composition and depression [44].

*Anxiety:* All studies investigating anxiety in children found that meeting three recommendations was associated with lower anxiety [49–52] and three studies found the same association in children meeting individual recommendations [49,51] or a combination [52].

*Resilience and self-esteem:* All studies to include resilience found an association; meeting the sleep recommendation [49], MVPA+ST [52] or a higher number of guidelines [51] were associated with higher resilience. Neither of the two studies assessing self-esteem found an association of movement behaviours with this outcome [44,53].

*Quality of life and general mental health:* Two studies to investigate quality of life did not find any significant associations of movement behaviours with this outcome in children [53,54], while the third found the more guidelines met, the higher the life satisfaction [55]. Meeting more guidelines [51,56] and specifically MVPA and ST were associated with better wellbeing [57].

The results are summarised in Table 3.

**Table 3. Number of studies in children (N = 27) finding positive, negative or no significant associations between movement behaviours and outcome, overall and according to analysis type.**

| Outcome | Analyses | Positive | None | Negative |
|---|---|---|---|---|
| Emotional problems n = 18 | Total | 14 | 6 | 4 |
| | Meet guidelines yes/no n = 12 | 8 | 4 | 2 |
| | CoDA n = 5 | 4 | 1 | 1 |
| | ISM n = 4 | 2 | 1 | 1 |
| Depression n = 7 | Total | 6 | 1 | 0 |
| | Meet guidelines yes/no n = 6 | 6 | 0 | 0 |
| | CoDA n = 1 | 0 | 1 | 0 |
| Anxiety n = 4 | Total | 4 | 0 | 0 |
| | Meet guidelines yes/no n = 4 | 4 | 0 | 0 |
| Resilience n = 3 | Meet guidelines yes/no | 3 | 0 | 0 |
| Self-esteem n = 2 | Total | 0 | 2 | 0 |
| | Meet guidelines yes/no n = 1 | 0 | 1 | 0 |
| | CoDA n = 1 | 0 | 1 | 0 |
| Quality of life n = 3 | Total | 1 | 2 | 0 |
| | Meet guidelines yes/no n = 2 | 1 | 1 | 0 |
| | CoDA n = 1 | 0 | 1 | 0 |
| Mental health n = 3 | Meet guidelines yes/no | 3 | 0 | 0 |

Note: CoDA; compositional data analysis, ISM; isotemporal substitution analysis. Some studies used more than one analysis technique and are therefore represented more than once within an outcome. Some studies are represented in more than one column for emotional problems because they used more than one measure for this outcome, i.e., internalising problems, externalising problems and total score or more than one analysis technique (details in S2 Table in S1 File).

**3.2.2 Adolescents.** *Emotional problems:* All studies found an association of movement behaviours with emotional problems in adolescents. Six of seven studies using guidelines found that meeting more recommendations was associated with lower emotional problems [37,41,42,58–60].

Of the three studies employing CoDA, one found higher sedentary and higher ST were associated with more emotional problems while higher sleep and LPA were associated with lower problems [61]. A second study also found higher sedentary time to be associated with higher problems [44]. The third study found higher sleep and MVPA were associated with lower emotional problems [62]. The two studies using ISM found replacing with sedentary/screen time with sleep or MVPA led to lower emotional problems [44,63].

*Depression:* Of the 23 studies including depression as an outcome, 21 found an association with movement behaviours. Twelve of 13 studies assessing meeting the guidelines found that meeting more recommendations was associated with lower depression [48,50,51,64–72]. Four studies found that meeting sleep individually was associated with lower depression [48,64,68,69] and five found that those meeting both sleep and ST had lower depression [52,65,66,69,70]. Meeting ST [48,64], MVPA [48,68], or a combination thereof [68,69] were also associated with lower depression. However one study found meeting only MVPA led to an increase in depression [70].

**Table 4. Number of studies in adolescents (N = 41) finding positive, negative or no significant associations between movement behaviours and outcome, overall and according to analysis type.**

| Outcome | Analyses | Positive | None | Negative |
|---|---|---|---|---|
| Emotional problems n = 11 | Total | 10 | 1 | 1 |
| | Meet guidelines yes/no (n = 7) | 6 | 1 | 1 |
| | CoDA (n = 3) | 3 | 0 | 0 |
| | ISM (n = 2) | 2 | 0 | 0 |
| Depression n = 23 | Total | 21 | 2 | 1 |
| | Meet guidelines yes/no (n = 13) | 13 | 0 | 1 |
| | CoDA (n = 5) | 4 | 1 | 0 |
| | ISM (n = 4) | 3 | 1 | 0 |
| | Latent profile/cluster (n = 2) | 2 | 0 | 0 |
| Anxiety n = 13 | Total | 13 | 0 | 1 |
| | Meet guidelines yes/no (n = 8) | 8 | 0 | 0 |
| | CoDA (n = 2) | 2 | 0 | 0 |
| | ISM (n = 3) | 3 | 0 | 1 |
| Resilience n = 5 | Total | 5 | 0 | 0 |
| | Meet guidelines yes/no (n = 2) | 2 | 0 | 0 |
| | CoDA (n = 1) | 1 | 0 | 0 |
| | ISM (n = 1) | 1 | 0 | 0 |
| | Latent profile (n = 1) | 1 | 0 | 0 |
| Self-esteem n = 4 | Total | 3 | 1 | 0 |
| | Meet guidelines yes/no (n = 1) | 1 | 0 | 0 |
| | CoDA (n = 1) | 0 | 1 | 0 |
| | ISM (n = 1) | 1 | 0 | 0 |
| | Latent profile (n = 1) | 1 | 0 | 0 |
| Stress n = 3 | Total | 3 | 0 | 0 |
| | Meet guidelines yes/no (n = 2) | 2 | 0 | 0 |
| | ISM (n = 1) | 1 | 0 | 0 |
| Quality of life/ Life satisfaction n = 4 | Total | 4 | 0 | 0 |
| | Meet guidelines yes/no (n = 3) | 3 | 0 | 0 |
| | CoDA (n = 1) | 1 | 0 | 0 |
| Mental health/ Distress/diagnosis n = 6 | Total | 6 | 0 | 0 |
| | Meet guidelines yes/no (n = 5) | 5 | 0 | 0 |
| | CoDA (n = 1) | 1 | 0 | 0 |
| Flourishing n = 6 | Total | 6 | 0 | 0 |
| | Meet guidelines yes/no (n = 1) | 1 | 0 | 0 |
| | CoDA (n = 1) | 1 | 0 | 0 |
| | ISM (n = 3) | 3 | 0 | 0 |
| | Latent profile (n = 1) | 1 | 0 | 0 |

*(Continued)*

 

**Table 4.** (Continued)

| Outcome | Analyses | Positive | None | Negative |
|---|---|---|---|---|
| Loneliness n = 1 | Meet guidelines yes/no (n = 1) | 1 | 0 | 0 |

Note: CoDA; compositional data analysis, ISM; isotemporal substitution analysis. Some studies used more than one analysis technique and are therefore represented more than once within an outcome. Some studies are represented in both the positive and negative columns because they found different effects for different movement behaviours.

Five studies employed CoDA: four found that higher sleep or higher LPA/MVPA were associated with lower depression [62,73–75], and two found that lower sedentary time [62,75] or lower ST [63,73] were also associated with lower depression. One study did not find any statistical effect [44]. Two studies used latent profile or cluster analysis and both found those with high MVPA and low ST showed lower depression [76,77]. Three of four studies using ISM found replacing ST with sleep led to lower depression [63,73,78]. Two additionally found replacing ST with MVPA [63,73] or MVPA with sleep [73,78] also led to lower depression.

*Anxiety:* Thirteen studies measured anxiety in adolescents. Seven of eight which analysed meeting the guidelines found that meeting more recommendations was associated with lower anxiety [50,51,65,66,69,71,79]. Four of those studies also found that those meeting both ST and sleep had lower anxiety [52,65,66,69]. One study employing CoDA found higher sleep, MVPA and lower ST to be associated with lower anxiety [74] and the other found higher LPA and lower sedentary time to be the case [75]. ISM analyses found that replacing ST or MVPA with sleep, or sedentary/ST with MVPA, led to lower anxiety [63,78,80].

*Resiliency:* Five studies assessed resiliency in adolescents. Meeting more guidelines was associated with higher resiliency [51]. A latent profile analysis found that those with any combination involving high MVPA or low ST had better resiliency [81] and CoDA analysis showed the same for higher sleep, MVPA and low ST [74]. The other study used ISM and found replacing ST with sleep or MVPA led to better resiliency [82].

*Self-esteem:* Four studies included self-esteem as an outcome. Meeting more guidelines was associated with higher self-esteem [83]. The study using CoDA found no association of movement behaviours on self-esteem in adolescents [44]. An ISM analysis found replacing ST with sleep or MVPA led to better self-esteem [82] and a latent profile analysis found those with higher MVPA and low ST had better self-esteem [81].

*Quality of life:* Four studies measured quality of life or life satisfaction. All looking at meeting the guidelines found that meeting more recommendations was associated with higher quality of life/satisfaction [55,59,84]. A CoDA analysis also found that higher sleep, higher MVPA and lower sedentary time were associated with higher life satisfaction [62].

*General mental health:* Five of the six studies on general mental health assessed meeting the guidelines. Of those, four found that meeting more recommendations was associated with better mental health [51,85–87]. Three found that meeting individual guidelines was also associated with better mental health [58,86,87]. The only study using CoDA found that higher ST was associated with higher distress [61].

*Flourishing:* Six studies measured flourishing. The one study looking at meeting the guidelines found meeting MVPA or sleep individually was associated with higher flourishing [88]. All studies using ISM found that replacing ST with sleep or MVPA or replacing sleep with MVPA resulted in higher flourishing [63,78,82]. A latent profile analysis found those with low ST showed higher flourishing [81].

*Stress:* Three studies measured stress: meeting the sleep recommendation was associated with lower stress [58] as well as all other combinations [83]. Replacing sedentary or LPA time with MVPA led to lower stress [80].

*Loneliness:* The only study to measure loneliness found that not meeting any recommendation was associated with higher loneliness [67].

### 3.2.3 Adults.

*Depression:* Twenty five of 29 studies found an association between depression and movement behaviours in adults. Eleven of 12 studies which assessed meeting the guidelines found meeting more recommendations was associated with lower depression [89–99], and nine found that meeting individual recommendations (MVPA, sedentary, sleep and/or ST), or combinations thereof, was associated with lower depression [90–97,100].

Four of 11 CoDA studies did not find any significant association of movement composition with depression in adults [20,101–103]. Five studies found that higher sedentary time and/or lower MVPA was associated with higher depression [104–110]. Two studies, interestingly, found that more sleep was associated with higher depression [106,107], but two found the opposite [108,110]. The only RCT used SEM and found that the intervention increased the composite activity-sleep score, which in turn led to lower depression [111].

Thirteen of 15 ISM analyses found a significant, positive statistical effect. Strikingly, 13 studies found that replacing sedentary time with MVPA resulted in lower depression [102–110,112–115]. Six studies also found lower depression by replacing sedentary time with sleep [102,104,106,108,113,114], but one found the opposite [107]. Replacing sleep with MVPA, sedentary time with LPA and LPA with MVPA were also beneficial for depression symptoms in a number of studies [102,105–110,112,113].

*Anxiety:* Nine of 13 studies found an association of movement behaviours with anxiety. Five of six studies analysing meeting the guidelines found an effect: four found meeting more recommendations was associated with lower anxiety [90,92,96,116] and four found individual and combination effects of meeting sleep, sedentary/ST and/or MVPA recommendations on lower anxiety [90,92,96,100].

Four studies employed CoDA, two of which found that higher sedentary time was associated with higher anxiety [105,110]. The other two studies did not find an association of movement composition with anxiety [20,101]. Three of the six studies using ISM did not find an effect of replacing behaviours on anxiety symptoms [20,101,112]. Three studies found replacing sedentary time with MVPA resulted in lower anxiety [21,105,110]. Replacing sedentary time with LPA had mixed results [21,105]. The RCT using SEM found that those with an increased composite activity-sleep score lowered their anxiety [111].

*Stress:* An association of movement behaviours with stress was found in 5/7 studies in adults. Both studies analysing meeting the guidelines found that meeting more recommendations was associated with lower stress [90,117].

Two of three studies using CoDA did not find a relationship between movement composition and stress and these same studies also did not find any effects using ISM [20,101]. However, two studies using ISM did find replacing sedentary time with sleep or MVPA led to lower stress levels [110,118]. The RCT using SEM found a higher composite activity-sleep score to lead to lower stress [111].

*General mental health:* All 14 studies with general mental health/mood/well-being as an outcome found an association with movement behaviours. Four of five studies investigating guidelines found meeting more recommendations was associated with better mood, mental health or lower distress [93,119–121].

Five of the six studies performing CoDA found higher MVPA was associated with higher well-being/affect/mental health [20,103,106,122,123]. Another study found higher sleep, lower sedentary, but also lower LPA, were associated with lower distress [124].

The ISM analyses produced more varied results, but 6/7 studies found that replacing sedentary time with another behaviour (MVPA: n = 4, sleep: n = 3, LPA: n = 2) led to better mental health, mood or lower distress [20,103,118,122,124,125]. Four studies also found that replacing sleep with MVPA led to better well-being/mood/mental health [20,103,106,122].

*Quality of life:* Five of seven studies found an association of movement behaviour with quality of life/happiness. Meeting more guidelines was associated with higher QoL [96]. When CoDA was used, one study found higher MVPA was

**Table 5. Number of studies in adults (N = 46) finding positive, negative or no significant associations between movement behaviours and outcome, overall and according to analysis type.**

| Outcome | Analyses | Positive | None | Negative |
|---|---|---|---|---|
| Depression n = 29 | Total | 25 | 4 | 2 |
| | Meet guidelines yes/no (n = 12) | 12 | 0 | 0 |
| | CoDA (n = 11) | 7 | 4 | 2 |
| | ISM (n = 15) | 13 | 2 | 2 |
| | Latent profile/SEM (n = 2) | 2 | 0 | 0 |
| Anxiety n = 13 | Total | 9 | 4 | 1 |
| | Meet guidelines yes/no (n = 6) | 5 | 1 | 0 |
| | CoDA (n = 4) | 2 | 2 | 0 |
| | ISM (n = 6) | 3 | 3 | 1 |
| | SEM (n = 1) | 1 | 0 | 0 |
| Stress n = 7 | Total | 5 | 2 | 0 |
| | Meet guidelines yes/no (n = 2) | 2 | 0 | 0 |
| | CoDA (n = 3) | 1 | 2 | 0 |
| | ISM (n = 3) | 1 | 2 | 0 |
| | SEM (n = 1) | 1 | 0 | 0 |
| Quality of life/ Life satisfaction n = 7 | Total | 5 | 2 | 0 |
| | Meet guidelines yes/no (n = 1) | 1 | 0 | 0 |
| | CoDA (n = 2) | 1 | 1 | 0 |
| | ISM (n = 4) | 2 | 2 | 0 |
| | SEM/PCA (n = 2) | 2 | 0 | 0 |
| Mental health/ Distress/Well-being/Mood n = 14 | Total | 14 | 0 | 1 |
| | Meet guidelines yes/no (n = 5) | 5 | 0 | 0 |
| | CoDA (n = 6) | 6 | 0 | 1 |
| | ISM (n = 7) | 7 | 0 | 0 |
| | Latent profile (n = 1) | 1 | 0 | 0 |
| Loneliness n = 3 | Total | 3 | 0 | 0 |
| | Meet guidelines yes/no (n = 1) | 1 | 0 | 0 |
| | CoDA (n = 2) | 2 | 0 | 0 |
| | ISM (n = 1) | 1 | 0 | 0 |
| Work-related mental health n = 3 | Total | 1 | 3 | 0 |
| | Meet guidelines yes/no (n = 1) | 0 | 1 | 0 |
| | CoDA (n = 2) | 1 | 1 | 0 |
| | ISM (n = 2) | 0 | 2 | 0 |
| PTSD n = 2 | Meet guidelines yes/no (n = 2) | 2 | 0 | 1 |

Note: CoDA; compositional data analysis, ISM; isotemporal substitution analysis, PCA; principal components analysis, SEM; structural equation modelling. Some studies used more than one analysis technique and are therefore represented more than once within an outcome. Some studies are represented in both the positive and negative columns because they found different effects for different movement behaviours.

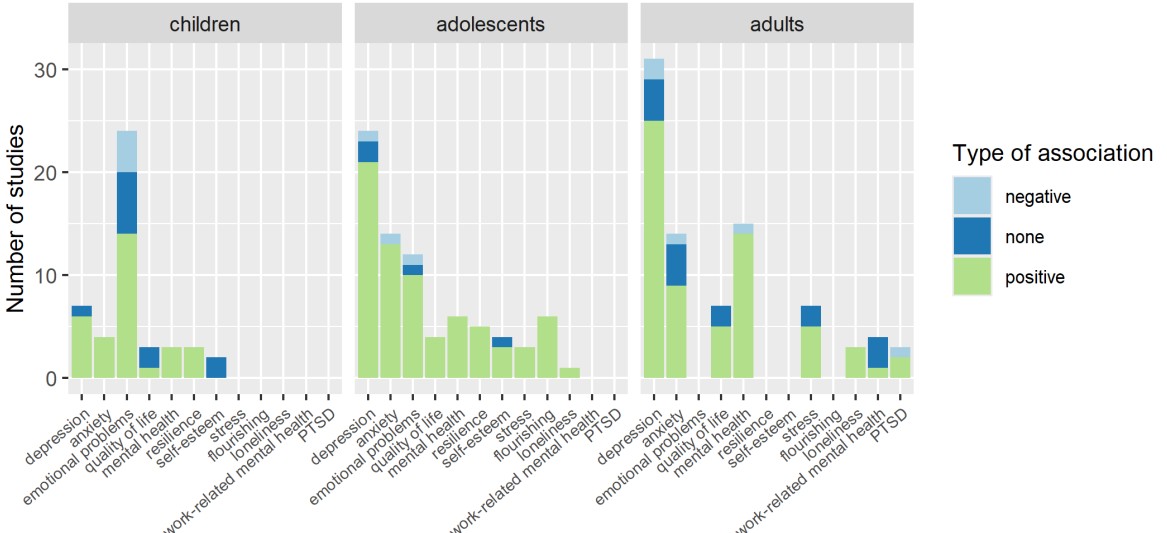

**Fig 2. Number of studies finding positive, negative or no significant associations between movement behaviours and outcome.**

associated with higher happiness [106] and one study found no significant effects [101]. The latter study also found no effect when using ISM, but two studies did find an effect: both found that replacing sedentary time with MVPA led to higher happiness/quality of life [106,126], one also found higher QoL when replacing sedentary with sleep [126] and the other found higher happiness when replacing sleep or LPA with MVPA.

Using SEM, the only RCT showed those with an increased composite activity-sleep score showed improved QoL [111]. Lastly, a PCA study found those with high MVPA/high sedentary or high LPA/high sedentary had better QoL [127].

**Work-related mental health:** Two of the three studies looking at shift-work problems, work engagement or burnout did not find any association of movement behaviour with these outcomes [20,119]. One study using CoDA found that higher sleep and lower sedentary time were associated with higher work engagement [124].

**Loneliness:** Meeting more guidelines was associated with lower loneliness [97]. Both CoDA analyses revealed higher MVPA was associated with lower loneliness in older adults [106,108]. Similarly, ISM analyses showed that increasing MVPA or reducing sedentary time also led to lower loneliness scores [106,108].

**PTSD:** Two studies assessed PTSD (military personnel/COVID-19) and both found meeting the sleep recommendation, in particular, was associated with fewer PTSD symptoms [93,128].

## 4. Discussion

This systematic review highlights the relationship between movement behaviours and mental health, especially in adolescents and adults. More movement within 24 hours was associated with at least one improved mental health outcome in the vast majority of studies. Furthermore, the fast-developing nature of the field is evident, with over half of the 103 studies in this review published since 2022.

The results in children are more mixed than in the other age categories. Emotional problems, which included internalising and externalising problems, were the most investigated outcomes and in the studies with a positive statistical effect, the overall finding seems to be that lower sedentary and lower screen time are associated with fewer emotional problems. However, half of the studies found no or negative effects, suggesting the relationship between movement and emotional

problems is not simple. Fewer studies investigated other mental health outcomes: most studies investigating depression, anxiety, resilience and general mental health found positive statistical effects but neither self-esteem study found an association and nor did two of three quality of life studies. Across outcomes 30% of studies found no significant association. These results are very similar to a scoping review in children and youth which found that 29–39% of studies showed no association between movement behaviours and mental health [24].

There are a number of potential explanations for the disparate results in children: First, it could be that children are more active and so there is less variability in the movement data. This is supported by findings that children are more likely to meet the movement guidelines than adolescents [58,129]. Second, it is possible studies with children are more sensitive to methodological differences, as suggested by McNeill et al. [130]. Third, perhaps measuring mental health is more difficult in children or children show less variability in mental health scores. However this seems unlikely as most studies employed outcome measures with strong psychometric properties (e.g., strengths and difficulties questionnaire [131] or the child behavior checklist [132]). Finally, movement behaviours are possibly not such an important driver of mental health at a younger age. These points are not mutually exclusive and likely a combination of factors is at play, but more research using controlled studies are needed to tease apart these possible explanations.

In adolescents 10/11 studies found a positive association with emotional problems, with all movement behaviours being involved. In general, the more recommendations from the 24-hour movement guidelines met, the lower the emotional problems. This was the same for depression, but with higher sleep, MVPA and lower ST in particular commonly associated with lower depression. The most common associations with lower anxiety were lower ST and higher sleep. Resiliency, self-esteem and flourishing were assessed by fewer studies, but for all, lower screen time seems to be important. Overall higher sleep and lower ST seem particularly relevant for adolescents.

Regarding general mental health and quality of life outcomes (n = 10), all studies found positive associations, but no clear conclusions can be drawn about specific movement behaviour patterns. This suggests that while certain movement behaviours or behaviour patterns may be more important for individual symptoms (e.g., higher sleep and lower ST for lower depression), an overall healthier movement behaviour routine is associated with better general mental well-being.

Adolescence is an important period for habit formation and for mental illness prevention. This age period sets the course for future health behaviours [133] and many mental disorders begin in adolescence [134]. It's a time when activity levels drop [58,129] and insufficient sleep in some countries is common to due late sleep onset in teenage years but a requirement to wake up early for school [135,136]. It is also the age group where the highest proportion of studies in this review found an association between movement and mental health (98%). Overall, this demographic is more susceptible to long-term consequences of poor physical and mental health. Therefore, adolescents are a crucial target group for healthy movement campaigns for mental illness prevention and healthy habit formation to carry into adulthood.

In adults, depression was by far the most common outcome with 29 studies, 25 of which found an association with movement. Across the findings, meeting more guidelines and reducing sedentary time were common associations with lower depression. The findings for anxiety were more mixed, with nine of 13 studies finding a positive statistical effect. Indeed, five of the six studies did find that meeting more recommendations was associated with lower anxiety, but other findings were much less consistent. The findings on stress are also not conducive to firm, specific conclusions, however lower sedentary time and higher sleep were found to be associated with lower stress in four of the seven studies (using different analysis techniques). In terms of general mental health and quality of life, all but two studies found a positive effect and multiple studies found a positive association with MVPA. Additionally, replacing sedentary time with another behaviour was often associated with better mental health. Overall, reducing sedentary time appears to be a common theme for better mental health in adults.

There was large heterogeneity in the adult study samples, with most sub-groups only investigated by one or two studies (over 45-year-olds, over 65s, office workers, inactive people, pre-school caregivers, nurses, military personnel, preconception/recently pregnant, visually impaired, cancer survivors, priests). Studies on the general population and university students/young adults had 10 and 11 studies respectively and can therefore be discussed in more detail.

Only one of the studies in the general population was on a representative sample [104], while many of the others were based on sub-samples of larger nationwide surveys. Interestingly all seven studies using ISM analyses found replacing sedentary time with another behaviour improved general mental health [125], affect [122] or reduced depression [104,107,109,115]. The studies using CoDA also found consistent results; increased sedentary time was associated with higher depression [104,107] and increased MVPA was associated with lower depression and better mental health [107,109,122,123]. In young adults, replacing sedentary time with other behaviours was associated with better mood [118], lower depression [102,103,110] and lower anxiety [21]. Additionally, there was a positive association of meeting the guidelines with anxiety [92,116], depression [92,94], PTSD [128], quality of life [96] and well-being [120]. These findings indicate there is some consistency in results across mental health outcomes. However due to the wide range of analysis techniques and mental health outcomes, more specific conclusions are difficult to draw. It is also apparent that much more research is needed with adult samples representative of the general population. Furthermore, recent studies suggest that some types of activity may have stronger impacts on mood and wellbeing than others. For example leisure-time, but not work-related physical activity, were related to happiness and stress, suggesting not only quantity, but also quality may be important [137]. Future studies could more closely assess which types of PA, sleep etc. within the movement composition are most beneficial. Future research is also necessary to explore the working mechanisms underlying how movement behaviours influence mental health. It would be interesting to explore which common or specific factors of movement behaviour compositions lead to improved mental health. Potentially, the working mechanisms known in psychotherapy research such as resource activation or mastery/coping/problem solving [138,139] could be working mechanisms also in the context of movement behaviours.

Only one study included in this review was a randomised controlled trial and therefore warrants specific mention. Duncan et al. [111] investigated the effects of a physical activity and sleep m-health intervention on quality of life and symptoms of depression, anxiety and stress in Australian physically inactive adults with poor sleep quality. They found that the intervention improved overall activity and sleep patterns, which in turn led to improvements in mental health outcomes. This corroborates the cross-sectional findings, suggesting not only an association, but that improving composite movement behaviours can improve mental health.

Across age groups, it is possible that methodological factors affect the likelihood of finding an association between movement behaviours and mental health. For example, some studies did not have mental health as a primary focus, but rather focused on, for example, physical health with mental health as a secondary outcome. However, these studies were not less likely to find an association (17/20 found a positive relationship). Likewise, studies with only a single mental health item rather than a standardised scale were also just as likely to find an association (7/7). Finally, perhaps studies with self/parent-reported, rather than objective, PA would be less likely to find an association, but this was also not the case (62/64 found a positive association between movement and mental health). Similarly when analysis technique is considered, there is not a particular statistical method which was less likely to find a positive association: 51/57 studies assessing meeting the guidelines found a positive association with mental health, 20/25 employing compositional data analysis, 25/32 using isotemporal substitution modelling and 7/7 using a form of cluster or latent profile analysis. Therefore the overall conclusion that healthier movement behaviours are associated with better mental health is seemingly robust across methodological and statistical techniques.

A previous review by Sampasa-Kanyinga et al. [22] on children and adolescents highlighted a number of suggestions for future research based on the literature up to that point, including longitudinal and experimental designs, robust measures of movement behaviours and validated measures of mental health. This review shows that since then the quality of

research has somewhat improved; advances in technology mean more and more studies employ objective measures of movement, validated mental health measures were also the norm and more robust analysis techniques which better control for the co-dependent nature of movement behaviours (CoDA) and assess where movement priorities should lie (ISM) were often used.

However, the certainty of evidence in this review was still very low and the findings have raised a number of research gaps which still remain and provide avenues for future research. It is clear that randomised controlled trials are still sorely lacking in this field and a focus on this in all age groups will contribute to a deeper understanding of any causal effects of movement behaviours on mental health. While the number of longitudinal studies has increased in recent years, the field would also benefit from a higher focus on this design to overcome many limitations of cross-sectional designs. The study samples in adult populations were very heterogenous and as such more studies with representative adult general populations would build baseline knowledge. Finally, the results cannot currently be generalised to African populations. While the included studies go beyond the typical WEIRD samples (Western, educated, industrialised, rich, democratic) with 17 countries represented, the vast majority were from three; Canada (n = 28), China (n = 28) and the USA (n = 17), and therefore generalisations across cultures cannot be systematically evaluated at this stage. Social and environmental differences in daily routines and accepted activity levels [140] may moderate the relationship between movement behaviours and mental health, as has previously been suggested for levels of physical activity [141].

## 4.1  Strengths and limitations

To our knowledge, this is the first systematic review to assess the relationship between 24-hour movement behaviours and mental health in adults and it significantly adds to the previous reviews in youth [22,24] as a large number of studies have been published since then. A comprehensive search process was employed, which was created in collaboration with an expert information specialist to capture all relevant studies. The review implemented a rigorous methodology following the PRISMA guidelines and using accepted methods of risk of bias and quality assessment.

There are limitations to the current review which should be taken into consideration. This review was intended to capture as much of the literature on movement behaviours and mental health as possible and therefore the heterogeneity of the included studies precluded a meta-analysis. Future studies with a more focused search criteria may be able to run meta-analyses on a more homogonous sample. However it is also worth reiterating here the dearth of experimental studies in this field, which would help to establish causal effects of movement behaviour on mental health. The cross-sectional nature of most of the studies meant that the certainty of evidence for the main outcomes was very low. Finally, with regard to the non-clinical target group of this review, there were two studies on cancer survivors, of whom some still received outpatient care [126,142]. While these participants did not meet our exclusion criteria of needing acute care, it may nevertheless be that these particular participants are a clinical sample but we decided to keep these papers in the review as the majority were non-clinical.

## 5.  Conclusion

Strikingly, almost all studies investigating the relationship between movement behaviours and mental health found a positive relationship. This was the case across measures and analysis techniques, but was a less robust finding in children. Investigating the relationship between 24-hour movement and mental health could have huge implications for public health. The 2019 global burden of disease study revealed that mental disorders are in the top 10 leading causes of burden worldwide [143]. It has been estimated that each year 38% of the EU population suffers a mental disorder, and disorders of the brain, including mental disorders, are the largest contributor to all cause morbidity burden in the EU [144]. Mental illness prevention and mental health promotion interventions are cost-effective and cost-saving methods of mental health care [145]. As such, researching guidelines that the general population can adopt in their daily lives could lead to economical methods of mental health maintenance that could hugely reduce the burden on psychological support services

and contribute to mental disorder prevention. Therefore our findings highlight the importance of developing campaigns encouraging people to meet the 24-hour movement guidelines. However, better quality evidence via experimental designs is also needed to control for the numerous confounding factors involved in order to tease apart the mechanisms by which movement behaviours affect mental health.

## Supporting information

**S1 File.** S1 Methods. S1 Table. PubMed search strategy. S1 Results. Further details on sub-groups and analysis types. S2 Table. summary of findings in children. S3 Table. summary of findings in adolescents. Supplementary table S4: summary of findings in adults.
(DOCX)

**S2 File.** Risk of bias – cross-sectional studies. Risk of bias – longitudinal studies. Risk of bias – RCT study. Risk of bias summary per outcome. Grade quality assessment.
(XLSX)

**S3 File.** PRISMA checklist.
(XLSX)

**S4 File.** All extracted data for included and excluded studies.
(XLSX)

## Acknowledgments

We thank Irma Klerings for her expertise in checking and improving the search strategies.

## Author contributions

**Conceptualization:** Rachel Dale, Thomas Probst.

**Data curation:** Rachel Dale, Teresa O'Rourke.

**Formal analysis:** Rachel Dale, Teresa O'Rourke.

**Investigation:** Rachel Dale, Barbara Nussbaumer-Streit.

**Methodology:** Rachel Dale, Teresa O'Rourke, Barbara Nussbaumer-Streit, Thomas Probst.

**Project administration:** Rachel Dale.

**Supervision:** Barbara Nussbaumer-Streit, Thomas Probst.

**Visualization:** Rachel Dale.

**Writing – original draft:** Rachel Dale.

**Writing – review & editing:** Rachel Dale, Teresa O'Rourke, Barbara Nussbaumer-Streit, Thomas Probst.

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
