## [Decision Letter · Decision Letter 0]

5 Mar 2025

PONE-D-24-5050224-hour movement behaviours and mental health in non-clinical populations; a systematic review.PLOS ONE

Dear Dr. Dale,

Thank you for submitting your manuscript to PLOS ONE. After careful consideration, we feel that it has merit but does not fully meet PLOS ONE’s publication criteria as it currently stands. Therefore, we invite you to submit a revised version of the manuscript that addresses the points raised during the review process.

 Please submit your revised manuscript by Apr 19 2025 11:59PM. If you will need more time than this to complete your revisions, please reply to this message or contact the journal office at plosone@plos.org . Please include the following items when submitting your revised manuscript:

We look forward to receiving your revised manuscript.

Kind regards,

Heather Macdonald, Ph.D

Academic Editor

PLOS ONE

2. In the online submission form you indicate that your data is not available for proprietary reasons and have provided a contact point for accessing this data. Please note that your current contact point is a co-author on this manuscript. According to our Data Policy, the contact point must not be an author on the manuscript and must be an institutional contact, ideally not an individual. Please revise your data statement to a non-author institutional point of contact, such as a data access or ethics committee, and send this to us via return email. Please also include contact information for the third party organization, and please include the full citation of where the data can be found.

3. As required by our policy on Data Availability, please ensure your manuscript or supplementary information includes the following:

Additional Editor Comments:

Thanks for your patience with the lengthy review time! In addition to the comments from the Reviewers, please consider the following comments. In particular, in light of other reviews I located (e.g., Kracht et al., 2024) it will be important for the authors to highlight the unique contributions of their review.

1. Overall, the manuscript would benefit from thorough copy editing since this is not done by the journal if the manuscript is accepted for publication. I noted punctuation and grammar errors throughout. Specifically, in the results, findings from cited studies should be in the past tense (e.g., “Eleven of 12 studies that assessed meeting the guidelines…”).

2. When referring to results of observational studies, please remove use of “effect” throughout the manuscript since only associations can be determined.

3. The abstract lacks detail. The word limit is 300 for PLOS One so I suggest including additional information such as more detail on the search strategy. In addition, last part of the sentence on lines 30-31 seems like a fragment. Also, the authors mention that this research area is in its infancy but yet there are so many studies – that seems a bit contradictory.

4. Given that this is a narrative review, conclusions regarding age group differences may need to be tempered. While I appreciate the authors’ effort to review the evidence across all ages, perhaps a more focused approach on just one age group is warranted. Related to this, conclusions about the heterogeneity across studies is not surprising given the broad scope of this review. The authors cite the SR by Janssen and colleagues when describing the search strategy. It’s interesting to note that Janssen and colleagues focused just on adults and on studies that use compositional data analysis – so a much more focused approach.

5. Intro, Line 78: Please provide a reference for the opening statement. Similarly, please provide references for the statement on line 85 (“…a number of researchers have now begun to investigate…”)

6. Intro, Lines 98-101: The wording of this sentence could be improved (i.e., Across 10 studies [were they all cross-sectional?] included in the review, better mental health among children and youth was observed among those who met all three recommendations as compared with those who did not meet recommendations. A dose-response gradient was also observed…). Also, please cite the systematic review by Wilhite et al. (https://doi.org/10.1093/aje/kwac212) as they also investigated the relationship between movement behaviours and mental health (among other outcomes). There is also a recent systematic review by Zhao et al. (https://doi.org/10.3389/fpubh.2024.1351972)

7. Line 105: Regarding the statement that no reviews on this topic have included adults, there is a recent umbrella review by Kracht and colleagues (https://doi.org/10.1186/s44167-024-00064-6) that should be cited, and the review by Rollo and colleagues covered the entire lifespan, as did a 2018 review by Grgic et al. (https://doi.org/10.1186/s12966-018-0691-3).

8. Intro, Line 107: Please use the same order for the populations as in the rest of the paper.

9. Methods – Eligibility criteria – the subheading numbers for each PICOS item should be at the next level heading (e.g., 2.2.1 for Population, 2.2.2 for Intervention/exposure etc.)

10. Please provide the age range for each population.

11. Methods – study designs: I am unclear what the authors mean by “…accounting for different combinations of recommendations met by participants in the analyses” – perhaps an example would help here?

12. The publication year information on Lines 168-69 can be removed since it is already in the results.

13. Line 180: Regarding percentage of each “gender” – I assume that most studies reported biological sex and not the social construct of gender, so please update accordingly.

14. For the longitudinal studies, it would help to know the duration of follow-up. On a related note, while intervention studies/RCTs will be helpful, it seems that more longitudinal studies are needed. This was also highlighted as an area of future research in the umbrella review by Kracht and colleagues.

15. The discussion repeats the results in some sections (e.g., lines 435-443).

Reviewers' comments:

Reviewer's Responses to Questions

**Comments to the Author**

1. Is the manuscript technically sound, and do the data support the conclusions?

Reviewer #1: Yes

Reviewer #2: Yes

2. Has the statistical analysis been performed appropriately and rigorously? 

Reviewer #1: N/A

Reviewer #2: Yes

3. Have the authors made all data underlying the findings in their manuscript fully available?

Reviewer #1: Yes

Reviewer #2: Yes

4. Is the manuscript presented in an intelligible fashion and written in standard English?

Reviewer #1: Yes

Reviewer #2: Yes

5. Review Comments to the Author

Reviewer #1: Thank you for the opportunity to review this submission. Some suggestions as follows:

- I recommend updating the keywords as they appear too similar to the title. Keywords should facilitate additional insights to the paper and expand its visibility

- pg 3 line 42: not just the amount but the type/quality/frequency impact someone's health

- pg 3, line 50: what are you trying to get across? Yes we have finite time during the day but are you indicating "finite time to complete all work/life/health expectations"? it's an incomplete thought

- pg 5, line 90-94: references needed to support your claims

- pg 6, line 128 is redundant (MVPA is already explained as an acronym on pg 3, line 55)

- pg 7, line 133: this is not a sentence. please rewrite.

- pg 7, line 137: why are these phrases underlined?

- why was the search frame limited to 10 years?

- since you exported to excel, how were duplicates removed? was this an automatic process?

- page 12 "emotional problems" needs to be rephrased. this isn't accurate and is negatively framing symptoms affiliated with mental health. suggest using "emotional disruptions" or something akin

- your subheadings are confusing. "emotional problems" shows up on page 12, line 244 and then again on page 14, line 277. what are the differences? can they not be consolidated?

- page 26, line 546. you need to rephrase this sentence because there are plenty of research studies on movement and mental health in adults - yours is focused on 24-hour movement in particular. additionally, the sentence written as-is, it's a runon sentence

- pg 26, line 554: you say you wanted to capture as much as possible, but you've limited the search to the last 10 years. why?

Reviewer #2: Dear Authors,

I would like to thank you for the opportunity to review your manuscript. Your study addresses an important and timely topic, examining the relationship between movement behaviors and mental health across different age groups. The paper provides a comprehensive review of existing literature and presents valuable findings that contribute to the growing body of research in this field.

Below, I provide detailed comments and suggestions that I believe will help enhance the clarity, methodological rigor, and impact of your manuscript. I hope these recommendations will be useful in refining your work.

Major Comments

The introduction should more precisely formulate the research problem so that the reader clearly understands the main hypothesis or research question. Currently, the rationale for the study is somewhat broad and could be more explicitly defined.

Some topics in the introduction change abruptly and without clear transitions. For example, after discussing physical health (lines 41–60), the text suddenly shifts to methodological analysis methods (lines 61–76) and then to mental health (lines 77–88).

The description of movement behavior analysis methods (CoDA, ISM, lines 61–76) is overly detailed.

The introduction presents data from studies on children and adolescents (lines 97–105), but their connection to adult mental health is not clearly established.

Since the article aims to examine associations across all age groups, it would be beneficial to balance the analysis of studies from different age groups and explicitly highlight their interconnections.

Although the literature search was conducted in PubMed, Scopus, and Embase (lines 156–169), there is no explanation for why other important databases, such as Web of Science, were not included.

It is mentioned that most studies were observational ("Due to the observational nature of most studies, the quality was rather low", line 218), but there is no detailed analysis of the main quality shortcomings (e.g., selection bias, differences in measurement methods, or lack of control variables).

The results indicate that certain movement behaviors are associated with better mental health, but there is no discussion of the potential mechanisms that could explain these associations.

The discussion section focuses more on data from adolescents and adults, while the findings from the children’s group are discussed only briefly, despite the fact that their results were more contradictory.

It is noted that the findings may not be generalizable to African populations, but there is a lack of a broader discussion on how cultural and social factors might influence the association between movement behaviors and mental health.

Although the studies vary significantly in methodology, a more in-depth discussion of how this heterogeneity may influence the findings is needed.

It would be beneficial to expand the discussion on the potential mechanisms explaining the relationship between movement behaviors and mental health.

Recent studies (e.g., Skurvydas et al., 2024, in BMC Public Health and PLOS ONE) highlight that leisure-time physical activity has a stronger impact on mood and well-being compared to work-related physical activity.

Overall, this study presents valuable insights into the relationship between movement behaviors and mental health, making a meaningful contribution to the field. However, addressing the aforementioned methodological and interpretative issues will significantly improve the manuscript’s clarity, coherence, and scientific rigor.

I hope these suggestions will be helpful in enhancing your paper. I appreciate your work and look forward to seeing the revised version.

Best regards,

6. PLOS authors have the option to publish the peer review history of their article (what does this mean? ). If published, this will include your full peer review and any attached files.

**Do you want your identity to be public for this peer review?** For information about this choice, including consent withdrawal, please see our Privacy Policy .

Reviewer #1: No

Reviewer #2: No

---

## [Author Response · Author response to Decision Letter 1]

17 Apr 2025

Please see file 'Response to Reviewers'.

Formatting has been edited according to the provided links.

2. In the online submission form you indicate that your data is not available for proprietary reasons and have provided a contact point for accessing this data. Please note that your current contact point is a co-author on this manuscript. According to our Data Policy, the contact point must not be an author on the manuscript and must be an institutional contact, ideally not an individual. Please revise your data statement to a non-author institutional point of contact, such as a data access or ethics committee, and send this to us via return email. Please also include contact information for the third party organization, and please include the full citation of where the data can be found.

All data can now be found in the supplementary materials.

3. As required by our policy on Data Availability, please ensure your manuscript or supplementary information includes the following:

This data is now available in the supplementary materials (supplementary file 4).

Additional Editor Comments:

Thanks for your patience with the lengthy review time! In addition to the comments from the Reviewers, please consider the following comments. In particular, in light of other reviews I located (e.g., Kracht et al., 2024) it will be important for the authors to highlight the unique contributions of their review.

Thank you for pointing this out. Our review is the first to focus on mental health, rather than multiple health outcomes, with adult populations included. We have included a sentence in the introduction with references to highlight that most reviews have focused on physical health (line 80):

“Previous reviews have assessed the association between 24-hour movement and general health outcomes (7,12–14).”

We have also edited the aims sections to clarify our contribution to the literature.

1. Overall, the manuscript would benefit from thorough copy editing since this is not done by the journal if the manuscript is accepted for publication. I noted punctuation and grammar errors throughout. Specifically, in the results, findings from cited studies should be in the past tense (e.g., “Eleven of 12 studies that assessed meeting the guidelines…”).

Thank you, a final proof read has been conducted.

2. When referring to results of observational studies, please remove use of “effect” throughout the manuscript since only associations can be determined.

Thank you, we have edited accordingly throughout the manuscript.

3. The abstract lacks detail. The word limit is 300 for PLOS One so I suggest including additional information such as more detail on the search strategy. In addition, last part of the sentence on lines 30-31 seems like a fragment. Also, the authors mention that this research area is in its infancy but yet there are so many studies – that seems a bit contradictory.

This is true! Indeed it was in its infancy when we started but the updated search revealed a lot of new studies. We have rephrased to reflect the new status and added more details:

“There is increasing evidence that daily movement behaviours are associated with mental health. However the research into the relationship between 24-hour-movement and mental health, particularly in adults, is still to be systematically reviewed.”

4. Given that this is a narrative review, conclusions regarding age group differences may need to be tempered. While I appreciate the authors’ effort to review the evidence across all ages, perhaps a more focused approach on just one age group is warranted. Related to this, conclusions about the heterogeneity across studies is not surprising given the broad scope of this review. The authors cite the SR by Janssen and colleagues when describing the search strategy. It’s interesting to note that Janssen and colleagues focused just on adults and on studies that use compositional data analysis – so a much more focused approach.

The sentence on age group differences has been re-worded in the abstract.

We agree that the heterogeneity is not surprising but we feel it is still important to raise the point to explain why no meta-analysis was conducted. We have removed it from the certainty of evidence sentence to avoid the connotation that this is a limitation.

It is common practice to use a previous systematic review to assess the success of a search strategy and our informatics specialist advised us to use one which focused on analysing the composition of movement behaviours as this was an important inclusion criterion for us. Naturally our search found a much broader range of studies, but the important point was that our search found all of the studies included in Janssen et al.

5. Intro, Line 78: Please provide a reference for the opening statement. Similarly, please provide references for the statement on line 85 (“…a number of researchers have now begun to investigate…”)

A reference has been added at the start, thank you. We have used the current results as evidence of the extent of research into movement + mental health since we are the first to summarise this evidence.

6. Intro, Lines 98-101: The wording of this sentence could be improved (i.e., Across 10 studies [were they all cross-sectional?] included in the review, better mental health among children and youth was observed among those who met all three recommendations as compared with those who did not meet recommendations. A dose-response gradient was also observed…). Also, please cite the systematic review by Wilhite et al. (https://doi.org/10.1093/aje/kwac212) as they also investigated the relationship between movement behaviours and mental health (among other outcomes). There is also a recent systematic review by Zhao et al. (https://doi.org/10.3389/fpubh.2024.1351972).

The wording has been edited as suggested. The suggested citations have also now been included in the introduction (lines 101-116).

7. Line 105: Regarding the statement that no reviews on this topic have included adults, there is a recent umbrella review by Kracht and colleagues (https://doi.org/10.1186/s44167-024-00064-6) that should be cited, and the review by Rollo and colleagues covered the entire lifespan, as did a 2018 review by Grgic et al. (https://doi.org/10.1186/s12966-018-0691-3).

While the above reviews are important, they all focus on physical health with some mental health included. We have reworded to emphasise our focus on mental health outcomes (lines 119-121).

8. Intro, Line 107: Please use the same order for the populations as in the rest of the paper.

Edited.

9. Methods – Eligibility criteria – the subheading numbers for each PICOS item should be at the next level heading (e.g., 2.2.1 for Population, 2.2.2 for Intervention/exposure etc.)

Edited.

10. Please provide the age range for each population.

Good point - Included on lines 140-142:

“The age range definitions can vary slightly from study to study (supplementary materials) however typically children were 12 or under, adolescents 13-18 and adults >18 years.”

11. Methods – study designs: I am unclear what the authors mean by “…accounting for different combinations of recommendations met by participants in the analyses” – perhaps an example would help here?

Thank you for pointing out this unclear statement. An example of latent class analysis has been included here.

12. The publication year information on Lines 168-69 can be removed since it is already in the results.

Removed.

13. Line 180: Regarding percentage of each “gender” – I assume that most studies reported biological sex and not the social construct of gender, so please update accordingly.

Indeed true thank you. Some but not all report sex rather than gender so we have added sex to the sentence.

14. For the longitudinal studies, it would help to know the duration of follow-up. On a related note, while intervention studies/RCTs will be helpful, it seems that more longitudinal studies are needed. This was also highlighted as an area of future research in the umbrella review by Kracht and colleagues.

A range has been added to the manuscript (lines 245-246) and the duration for each study has been added to the respective notes in supplementary file 4.

Indeed longitudinal studies are very important and we have now highlighted this in the discussion (lines 586-588).

15. The discussion repeats the results in some sections (e.g., lines 435-443).

This is true, however we always attempted to summarise rather than repeat and the results highlighted in the discussion are to lead into a further discussion of the interpretation. Therefore we have chosen to leave this as is.

Reviewers' comments:

Reviewer's Responses to Questions

Comments to the Author

1. Is the manuscript technically sound, and do the data support the conclusions?

Reviewer #1: Yes

Reviewer #2: Yes

2. Has the statistical analysis been performed appropriately and rigorously?

Reviewer #1: N/A

Reviewer #2: Yes

3. Have the authors made all data underlying the findings in their manuscript fully available?

Reviewer #1: Yes

Reviewer #2: Yes

4. Is the manuscript presented in an intelligible fashion and written in standard English?

Reviewer #1: Yes

Reviewer #2: Yes

5. Review Comments to the Author

Reviewer #1: Thank you for the opportunity to review this submission. Some suggestions as follows:

Thank you for the helpful suggestions!

- I recommend updating the keywords as they appear too similar to the title. Keywords should facilitate additional insights to the paper and expand its visibility

The keywords were chosen with support of the information specialist to be similar to, but not the same as, the title. However indeed some were the same as the title so we’ve now edited them to be somewhat broader.

- pg 3 line 42: not just the amount but the type/quality/frequency impact someone's health

Indeed true, we have edited accordingly. We have also added a statement in the discussion regarding this point (lines 541-545):

“Furthermore, recent studies suggest that some types of activity may have stronger impacts on mood and wellbeing than others. For example leisure-time, but not work-related physical activity, were related to happiness and stress, suggesting not only quantity, but also quality may be important (138). Future studies could more closely assess which types of PA, sleep etc. within the movement composition are most beneficial.”

- pg 3, line 50: what are you trying to get across? Yes we have finite time during the day but are you indicating "finite time to complete all work/life/health expectations"? it's an incomplete thought

The thought is completed in the following sentence (Consequently…). The sentences are split for emphasis, a stylistic writing choice we hope the reviewer will accept us leaving.

- pg 5, line 90-94: references needed to support your claims

Thank you, references have been added (line 98).

- pg 6, line 128 is redundant (MVPA is already explained as an acronym on pg 3, line 55)

True, thank you! Full terms deleted.

- pg 7, line 133: this is not a sentence. please rewrite.

The sentence has been edited.

- pg 7, line 137: why are these phrases underlined?

Because it was very important to highlight this aspect of the inclusion criteria. We have now italicised them rather than underline.

- why was the search frame limited to 10 years?

It was not limited but rather this was the range of the included studies. We realise this does not belong here and has been included in the results instead.

- since you exported to excel, how were duplicates removed? was this an automatic process?

As stated on line 192-193, duplicates were removed in a reference manager, not excel. Data extraction took place in excel (line 198).

- page 12 "emotional problems" needs to be rephrased. this isn't accurate and is negatively framing symptoms affiliated with mental health. suggest using "emotional disruptions" or something akin

“Emotional problems” is a common term utilised in the literature and therefore for conformity with common terminology we have chosen to keep it in. However we agree it would benefit from a definition and have added one (lines 271-272). 

---

## [Decision Letter · Decision Letter 1]

29 Apr 2025

PONE-D-24-50502R124-hour movement behaviours and mental health in non-clinical populations; a systematic review.PLOS ONE

Dear Dr. Dale,

Thank you for submitting your manuscript to PLOS ONE. After careful consideration, we feel that it has merit but does not fully meet PLOS ONE’s publication criteria as it currently stands. Therefore, we invite you to submit a revised version of the manuscript that addresses the points raised during the review process.

Thank you for addressing my comments and those of the 2 reviewers. My only remaining comment is that although the authors modified use of "effects" when referring to their own findings (except on line 36 in the abstract - "effects" needs to be changed), the same needs to be done when citing results of previous observational studies. For example, for the statement on lines 49-51 - were the cited reviews able to confirm evidence of "reciprocal effects" or just associations? Please check use of effect throughout the manuscript.

We look forward to receiving your revised manuscript.

Kind regards,

Heather Macdonald, Ph.D

Academic Editor

PLOS ONE

Journal Requirements:

Additional Editor Comments:

Thank you for addressing my comments and those of the 2 reviewers. My only remaining comment is that although the authors modified use of "effects" when referring to their own findings (except on line 36 in the abstract - "effects" needs to be changed), the same needs to be done when citing results of previous observational studies. For example, for the statement on lines 49-51 - were the cited reviews able to confirm evidence of "reciprocal effects" or just associations? Please check use of effect throughout the manuscript.

Reviewers' comments:

Reviewer's Responses to Questions

**Comments to the Author**

1. If the authors have adequately addressed your comments raised in a previous round of review and you feel that this manuscript is now acceptable for publication, you may indicate that here to bypass the “Comments to the Author” section, enter your conflict of interest statement in the “Confidential to Editor” section, and submit your "Accept" recommendation.

Reviewer #2: All comments have been addressed

2. Is the manuscript technically sound, and do the data support the conclusions?

Reviewer #2: Yes

3. Has the statistical analysis been performed appropriately and rigorously? 

Reviewer #2: N/A

4. Have the authors made all data underlying the findings in their manuscript fully available?

Reviewer #2: Yes

5. Is the manuscript presented in an intelligible fashion and written in standard English?

Reviewer #2: Yes

6. Review Comments to the Author

Reviewer #2: Thank you for the opportunity to review the revised version of this manuscript. The authors have significantly improved the paper, presenting a well-structured, clearly written systematic review that adheres to PRISMA guidelines. The manuscript covers a broad and up-to-date range of literature.

The thorough and thoughtful discussion section addresses consistent and contradictory findings across different age groups.

7. PLOS authors have the option to publish the peer review history of their article (what does this mean? ). If published, this will include your full peer review and any attached files.

**Do you want your identity to be public for this peer review?** For information about this choice, including consent withdrawal, please see our Privacy Policy .

Reviewer #2: No

---

## [Editor Report · Decision Letter 2]

13 May 2025

24-hour movement behaviours and mental health in non-clinical populations; a systematic review.

PONE-D-24-50502R2

Dear Dr. Dale,

We’re pleased to inform you that your manuscript has been judged scientifically suitable for publication and will be formally accepted for publication once it meets all outstanding technical requirements.

Kind regards,

Heather Macdonald, Ph.D

Academic Editor

PLOS ONE
---

## [Editor Report · Acceptance letter]

PONE-D-24-50502R2

PLOS ONE

Dear Dr. Dale,

I'm pleased to inform you that your manuscript has been deemed suitable for publication in PLOS ONE. Congratulations! Your manuscript is now being handed over to our production team.

Kind regards,

on behalf of

Dr. Heather Macdonald

Academic Editor

PLOS ONE